

# An integrated route planning approach for driverless vehicle delivery system

Tianyang Li[1,2], Zhangyi He[1] and Yanling Wu[3]

[1] Computer Science, Northeast Electric Power University, Jilin, Jilin, China
[2] Jiangxi New Energy Technology Institute, Xinyu, Jiangxi, China
[3] Hebei Hanguang Industy Co., Ltd, Handan, Hebei, China

## ABSTRACT

With the rapid growth of express delivery in urban areas, the use of driverless vehicles as an alternative to traditional human delivery can reduce costs and improve efficiency. The route planning of driverless vehicles is crucial in realizing autonomous navigation, which improves the working level and ensures improvements in efficiency. However, it is difficult to reasonably organize the real-time delivery, taking into account several factors that influence the planning of routes, such as load capabilities, power limits and traffic conditions. To deal with this concern, we propose an integrated approach including a multistage model and improved genetic algorithm to obtain the optimal delivery plan for driverless vehicles. The experimental results in an urban scenario with a realistic delivery service show the superiority of our proposition in the delivery efficiency.

# INTRODUCTION

Driverless vehicles, also known as autonomous vehicles, are broadly defined as driverless robotic vehicles (*Kaur & Rampersad, 2018*). They can be used in urban areas, industrial parks and various terminal distribution scenarios that require short-distance logistics, which can effectively reduce transportation costs and improve the efficiency of logistics industry. Benefiting from the advance of artificial intelligence and new energy technology, the using of driverless vehicles, especially electric driverless vehicles (EDVs) will make delivery system more clean, efficiency and security (*Xie et al., 2017*; *Feng et al., 2021*). Our goal is to investigate the route planning approach for Driverless Vehicle Deliver System (DVDS) to improve distribution efficiency of the "last-kilometer" in an urban scenario.

In recent years, with the rise of online commerce, the logistics industry has developed rapidly. However, the current delivery system is still based on man power to deal with a large number of packages. There is a bottleneck on delivery efficiency and service quality on the "last-kilometer". Therefore, the DVDS becomes a promising way to replace manual distribution, and achieve costs reduction and efficiency improvements. That may upgrade the logistics industry with a new business model and further promote the development of online commerce.

Despite the advantages of DVDS, there are still some challenges for the improvement of delivering efficiency. These challenges mainly come from the remnant of batteries, payload, weather, traffic conditions, etc. The key to improve the delivering efficiency relies

Corresponding author
Zhangyi He, 626372333@qq.com

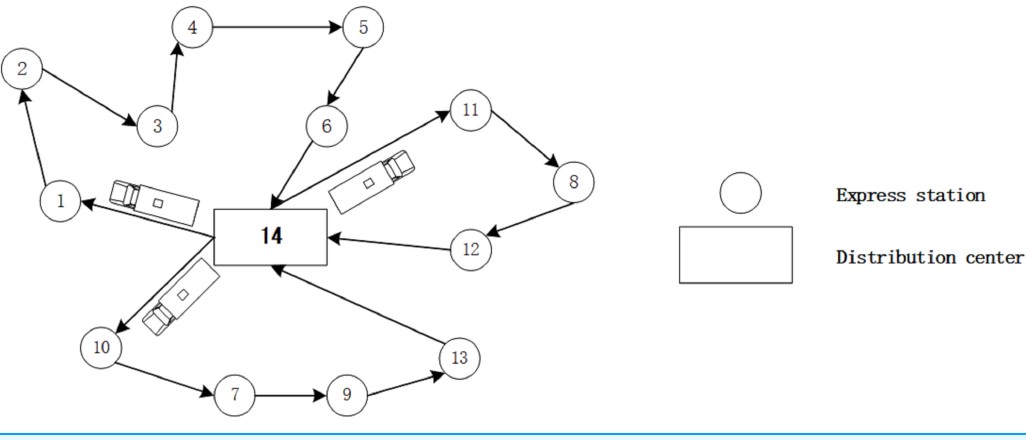

**Figure 1 An example of a distribution routes.**

on the reasonable arrangement of distribution routes between the Distribution Center (DC) to each Express Station (ES) or to the end user. In addition, for most logistics companies, their variable costs and delivery efficiency are mainly related to the specific distribution path selected for vehicles. Obviously, providing an effective delivery path that considers all the factors including remnant of batteries, payload, weather, traffic conditions and can assist in saving cost and improving efficiency is important. Therefore, an effective route planning approach for DVDS is indispensable for the sustainable development of the logistic industry.

Typically, there are many mathematical models for solving the route planning problem, such as the model of heterogeneous Fleet Vehicle Routing Problem (HFVRP), traditional Vehicle Routing Problem (VRP) and Dynamic Vehicle Routing Problem (DVRP). Here HFVRP refers to the fact that the whole distribution task should be completed by different types of vehicles and the number and cost of each type of vehicle may be different. DVRP is a VRP that satisfies certain dynamic constraints. However, considering the use of multiple EDVs to deliver packages in a real world context, many factors, such as the payload, remaining battery capabilities, road conditions and delivery priorities, should be taken into account when solving the problem. The traditional model lacks the expression of these crucial factors. For example, the HFVRP model doesn't consider the dynamic traffic situation and the DVRP model doesn't consider the different characteristics of multiple vehicles.

Figure 1 shows an example of DVDS with multiple driverless vehicles. There are three EDVs for the DC to deliver packages to ESs. Allocating ESs for each EDV is the first and foremost problem to be solved in the consideration of the rated payload and remaining battery capabilities. That means, with the target of maximum delivery quantity, each EDV should deliver more ESs within the distance based on its remaining capacity of batteries. When suitable ESs are identified for each EDV, the route of delivery should be re-planned according to the road condition. Obviously, this problem cannot be explained by a simple model, which makes the efficient solving of this problem more difficult. This may be an obstacle for the development of delivery industry.

Currently there is no comprehensive model that describes DVDS in the literature. The lack of literature precedent raises two questions: (i) How might these key factors be described in a uniform model? (ii) How to solve the uniform model effectively?

Therefore, an integrated approach including a multistage model and an improved GA (genetic algorithm) is proposed to describe and solve the route planning problem for DVDS with multiple EDVs.

The main contributions of this article may be threefold:

(i) designs a multistage model to describe the route planning problem with multiple EDVs;

(ii) proposes a GA-based multi-stage optimization strategy to solve the multistage model;

(iii) uses the data set from a real-world case to illustrate the application of proposed integrated approach, which may provide insights for delivery industry.

The article is structured as follows. "Related Works" section introduces the related literature. "Problem Definition" section describes the model entity, multistage model, assumptions and problem definition. Then in "Improved GA for the Multistage Model" section, we introduce the improved GA and its application with the multistage model. "Experiment and Result" section validate the proposed approach with a real-world case. Our conclusions, research limitations and further developments are presented in the last section.

## RELATED WORKS

According to a traffic technical report in the United States, 94% of road traffic accidents are caused by human error, such as fatigue driving and drunk driving (*Yurtsever et al., 2020*). If driverless cars replace conventional cars in the future, problems caused by human driving errors may be reduced. Results obtained from questionnaire survey in Germany have shown that autonomous delivery vehicles are believed to have the potential to revolutionize last-mile delivery in a more sustainable and customer-focused way *Kapser & Abdelrahman (2020)*.

Moreover, driverless vehicles also have many advantages. They can alleviate congestion and reduce exhaust emissions through intelligent travel strategies (*Thurner, Fursov & Nefedova, 2022*) and new energy technology. Therefore, there are lots of literature on the various areas involved in driverless driving (including drones) for delivery. For example, the new Air and Ground Autonomous Vehicle that has lower energy consumption and carbon dioxide emission when delivering goods was discussed (*Figliozzi, 2020*). Autonomous vehicles travel on circular networks for pickup and delivery was also studied (*Trotta et al., 2022*). Moreover, vehicle routing problem with drones (*Brunner et al., 2019*; *Lei, Cui & Li, 2022*), such as the Drone-as-a-Service (DaaS) (*Shahzaad et al., 2019*) and Swarm-based Drone-as-a-Service (SDaaS) (*Alkouz, Bouguettaya & Mistry, 2020*; *Alkouz & Bouguettaya, 2021*) were discussed. These studies have wildly investigated the potential of using driverless vehicles to delivery.

Obviously, driverless vehicles have got lots of attention despite they have not been put into use on a large scale. However, with the maturity of relevant technologies in the future,

**Table 1 Studies with different models.**

| Models | Algorithms |
|---|---|
| HFVRP | A developed Tabu search algorithm (*Meliani et al., 2019*). |
|  | Exact algorithms based on logic-based benders decomposition and a variant (*Fachini & Armentano, 2020*). |
|  | a hybrid meta-heuristic approach based on a developed Tabu search (TS) algorithm (*Meliani et al., 2022*). |
| VRP | Genetic algorithm combined with ant colony optimization algorithm (*Lesch et al., 2021*). |
|  | Ant colony algorithm combined to firefly algorithm (*Goel & Maini, 2018*). |
|  | Clustering algorithm is combined with ant colony algorithm (*Chen, 2020*). |
|  | Hybrid algorithm with neighborhood reduction (*He & Hao, 2022*). |
| DVRP | Best-cost route crossover (BCRC) based on genetic algorithm (*Alwabli & Kostanic, 2020*). |
|  | Evolutionary algorithms (EAs) (*Sabar et al., 2019*). |
|  | Discrete particle swarm optimization algorithm (*Daely et al., 2021*). |
|  | A meta-heuristic algorithm of requirement coverage diversity based on ant colony algorithm framework (*Xiang et al., 2020*). |

it is believed that they will be put into commercial use soon. Therefore, the research on driverless vehicle distribution is of great significance.

At the same time, many scholars have conducted multi-level research on vehicle routing in delivery distribution problem. Some literature have studied this problem with different models and presented many novel algorithms. Table 1 shows more details about these studies.

As we all know, Tabu search (TS), artificial bee colony (ABC), ant colony optimization (ACO) and particle swarm optimization (PSO) are also meta-heuristic algorithms like genetic algorithms. Obviously, these algorithms can overcome slow convergence and excessive randomness for effectively handling periodic demands, but still require a tremendous amount of hand labour to adjust mode for coping with the random surge demands properly. When the iteration reaches a certain level, the genetic algorithm and the above algorithms will converge prematurely due to the loss of population diversity. But we can know that the effectiveness of GAs extremely depends on genetic operators, we can achieve the goal of enriching population diversity by improving the genetic operators, so we proposed an improved mutation operator to help achieve this goal.

Meanwhile, according to above literature, we can find that evolutionary algorithms and swarm intelligence algorithm are widely used to a single model for solving VRP with certain circumstances. For example, (*Meliani et al., 2022*) uses TS algorithm to allocate loading plans for multiple vehicles under the condition that loading constraints are met. (*Goel & Maini, 2018*) uses firefly algorithm to achieve the overall goal of minimizing the driving distance while minimizing the number of vehicles required for delivery. (*Xiang et al., 2020*) is based on the ant colony algorithm framework to effectively respond to customers' dynamic requests. However, for considering more factors of DVDS, a single model in not enough and existing methods may be fail to solve the more complicated problem effectively. And we know that genetic algorithm can solve the vehicle routing problem and its deformation very well and play a cohesive role among the models with

chromosome as the carrier. Therefore, we propose a multistage model combined to an improved genetic algorithm to solve the distribution problem of DVDS efficiently.

# PROBLEM DEFINITION

In this section, we will introduce the proposed multistage model and define the problem in detail.

## Definition of entities

Firstly, five entities involved in the multistage model are introduced as follows.

(i) *DC:* Each DC can be represented as a triplet, $DC = <dcl, edvn, on>$, where *dcl* represents the geographical location of DC located by GPS, *edvn* represents the number of EDV in DC and *on* represents the number of orders received by logistics companies from customers. DC is the starting node of the DVDS transportation network. Companies design reasonable distribution routes based on ESs that need to be distributed and then select the appropriate EDVs for delivery.

(ii) *EDV:* Each EDV can represent a quad, $EDV = <vt, edvd, lw, lv>$, where *vt* represents the vehicle type, *edvd* represents the distance that the EDV can drive under the remaining battery capacity, *lw* represents the loading weight of EDV, *lv* represents the loading volume of EDV. The EDV types can be roughly divided into small, medium and large and they have different load capacities. Big EDV has a large load and a long distance to travel, so more ESs or farther ES can be arranged for distribution.

(iii) *ES:* Each ES can be expressed as a binary group, $ES = <esl, t>$, where *esl* refers to the geographical location of ES located by GPS and *t* refers to the type of ES, which mainly includes express station and delivery locker. For each delivery, all the packages stored in the ES will be delivered together.

(iv) *PA:* The package delivered by DVDS from DC to ES and can be expressed as a quintuple, $PA = <n, pal, es, pw, pv>$, where *n* represents the specific package number of each PA, *pal* represents the receiving place of PA, *es* represents the ES where the package is deposited, *pw* represents the weight of the package and pv represents the volume of the package. All this information is an important basis for package delivery.

(v) *TN:* The transportation network of a DVDS, which is a key factor in logistics and transportation, can be represented as a triplet, $TN = <s, d, c>$, where *s* represents all nodes in the network (including ES and DC), *d* represents the distance between all nodes and *c* represents the road congestion coefficient between all nodes. We can use $d_{ij}$ to represents the distance between station i and station j and use $c_{ij}$ to represents the road congestion coefficient between station i and station j. In this network, the paths between nodes are bidirectional and symmetric.

## The multistage model

Based on traditional VPR models and defined entities, we further proposed an multistage model to characterize the route planning problem of DVDS.

In general, the multistage model has three stages. The first stage is HFVRP model, which is mainly for ESs allocation; the second stage is VRP model, which is mainly to generate basic distribution routes; the third stage is the DVRP model, which is responsible for optimizing the distribution route.

(i) *Package Arrangement Model:* DC has several different types of EDV available for delivery and each vehicle has different battery power and load capacity. Assuming that various types of EDVs are fully charged, there are the following rules: the battery capacity of EDV is positively correlated with the vehicle loading capacity, *i.e.*, the farther the EDV travels, the more packages it can carry. This model mainly arranges EDV's driving routes according to the driving distance of EDV. For describing this problem, we employ the HFVRP that is used to find a fine-grained arrangement plan.

(ii) *Static Route Planning Model:* With the shared information including EDVs status and road condition, the optimal driving path between ESs (or end users) should be calculated. In this part, the driving route of each EDV can be obtained through solving the Static VRP model in a relatively short time.

(iii) *Dynamic Route Planning Model:* There may be uncertain factors in the distribution process, such as rush hour, special weather, traffic control and other situations, which may cause road congestion. If the EDV still follows the route derived from static path planning and drives into a congested road, it will increase the travel time for delivery, which ultimately affects the efficiency of delivery and usrs' satisfaction. Hence, a dynamic route planning model combined with Global Positioning System (GPS) is used to obtain the traffic situation between each ES and path planning, which enables the delivery path to be reasonably planned on the basis of new realistic conditions. When each EDV finishes the delivery for a certain ES and is about to drive to the next station, if congestion is found in the next road section, it will re-plan the route according to the current road conditions, so as to complete the delivery in the shortest time with small delivery distance. Since there are dynamic constraints, we can consider this problem as DVRP.

These three models are independent and correlated. Any single of them is not able to describe the VRP problem for DVDS. Consequently, multistage model is proper solution. For further detailing the formulation of this multistage model, there are several assumptions that are needed to introduce.

## Assumption of the multistage model
### Assumptions about DC:

(i) Only consider the case of single DC for each delivery routing planning.
(ii) The time for EDV to load packages in DC and unload packages in each ES are not considered, but the driving distance and driving time of EDV with its remnant power of batteries are mainly considered.

**Assumptions about EDV:**

(i) Each route is planed for only one EDV.

(ii) Ignore the power loss generated by each EDV when it stops or waits in congested roads.

(iii) Temporary faults on EDV or delivery errors are not considered during delivery.

**Assumptions about delivery routes:**

(i) The path between any two ESs can be regarded as an undirected arc, *i.e.*, an EDV can drive from $ES_i$ to $ES_j$ or from $ES_j$ to $ES_i$.

(ii) The path between any two ESs is bidirectional but the congestion coefficient only has one-way weight, *i.e.*, the coefficient matrix is a symmetric matrix.

Our focus is to plan the route of EDVs. Under the above restrictions and assumptions, we should make reasonable distribution arrangements.

## Problem formulation

The weighted network graph is used to describe the delivery network. Given a simple undirected graph $G = <v, e, c>$ with $N + 1$ nodes, where $v = \{v_1, v_2, \ldots, v_n, v_{n+1}\}$ is the node set in the graph. $v_{n+1}$ represents DC and the other nodes $(v_1 -> v_n)$ present ESs. The $e$ is the edge set of the graph and $c$ is the traffic congestion coefficient matrix. Other symbols are described as follows: $K$ is the number of EDV, $q_k$ represents the package quantity (including loading volume and loading weight) to be loaded by the *kth* EDV in DC and $Q_k$ refers to the rated maximum loading capacity of the *kth* EDV (also including loading volume and loading weight). $D_k$ represents the maximum driving distance of the *kth* EDV and $dd_k$ denotes the driving distance traveled by the *kth* EDV to complete all deliveries. $T_k$ represents the time taken for the *kth* EDV to complete all deliveries. $x_{ijk}$ refers to whether the *kth* EDV travels from $v_i$ to $v_j$, which is a variable of 0 and 1.

Aiming at minimizing the total route length and delivery time of delivery vehicles, we design a uniform mathematical model to describe the multistage model. The mathematical model established in this paper is as follows:

$$\min Z = \sum_{i=1}^{K} (dd_k + T_k) \tag{1}$$

$$s.t. \quad \sum_{k=1}^{K} \sum_{i=1}^{N+1} x_{ijk} = 1 \quad (j = 1, 2, \ldots, N) \tag{2}$$

$$\sum_{k=1}^{K} \sum_{j=1}^{N+1} x_{ijk} = 1 \quad (i = 1, 2, \ldots, N) \tag{3}$$

$$\sum_{j=1}^{N} x_{N+1jk} = \sum_{i=1}^{N} x_{iN+1k} <= 1, \quad k \in \{1, 2, \ldots, K\} \tag{4}$$

$$\sum_{k=1}^{K}\sum_{j=1}^{N} x_{N+1jk} <= K \tag{5}$$

$$Q_k - q_k >= 0, \quad k \in \{1, 2, \ldots, K\} \tag{6}$$

$$D_k - dd_k >= 0, \quad k \in \{1, 2, \ldots, K\} \tag{7}$$

$$x_{ijk} = \begin{cases} 1, & if \ \ the \ kth \ EDV \ delivers from \ v_i \ to \ v_j \\ 0, & otherwise \end{cases} \tag{8}$$

In the above model, the objective function (1) considers minimizing the length of delivery path and delivery time. Constraints (2) and (3) indicate that each ES's packages can only be delivered by one EDV, *i.e.*, there is no such thing as multiple vehicles delivering packages to the same ES. Constraint (4) means that EDV no longer return to DC to pick up packages when performing the delivery service but pick up all packages that need to be delivered at the beginning of EDV's delivery. Constraint (5) indicates that the number of EDVs departing from DC doesn't exceed the number of EDVs available for transport services. Constraint (6) refers to that the package quantity delivered by each EDV is less than the maximum loading capacity of the EDV, which includes loading weight and loading volume. Constraint (7) indicates that the remaining driving distance of each EDV is greater than the distance to be traveled for the distribution of all packages. Constraint (8) indicates that for the $k - th$ EDV, if it drives from $ES_i$ to $ES_j$, then $x_{ijk} = 1$, otherwise $x_{ijk} = 0$.

As we can see that the uniform mathematical model is more complicated and has a large-scale solution space. For obtaining a routing planing for all EVDs, there are at least three times of optimization of the objective function with different constraints. Of course, an effective optimization algorithm is needed. It is well known that GA can be used to all kinds of optimization problem. But it still has its own drawbacks, such as pre-mature and easily trapped into local optimization, especially with the large-scale solution space. Consequently, we further improve the GA to effectively solve the uniform model with different situations.

## IMPROVED GA FOR THE MULTISTAGE MODEL

In this section, we will use the improved genetic algorithm to solve the proposed uniform model. Firstly, the improvements of genetic algorithm and the application of it in different stages are detailed.

### Improved GA

Compared with other heuristic algorithms searching from a single point, GA searches from multiple points at the same time. However, traditional GA is easy to pre-mature when solving large-scale problems and cannot solve the uniform model effectively, so the optimal results can be efficiently obtained only by improving the GA.

GA usually consists of four stages: initial population, chromosome evaluation selection, crossover and mutation. The latter three stages need many iterations, then the new

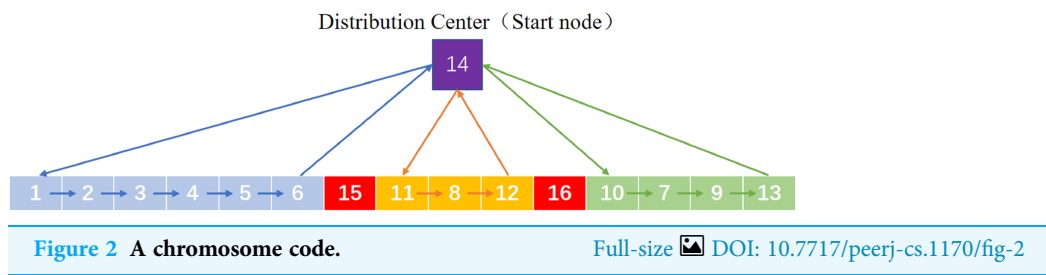

**Figure 2  A chromosome code.**

population will be better than the previous generation and gradually approach the global optimal solution. Next, the improvements of GA will be introduced below.

Encoding Settings: Genetic coding mainly adopts the set of natural number to represent the driving routes of $K$ EDV, numbers $1, 2, \ldots, N$ represents the ESs to be accessed by $K$ EDVs. In order to represent $K$ routes with one chromosome, $K - 1$ interruption points are added for route segmentation and these points are $N + 2, \ldots, N + K$. Each interruption point represents DC, so that the chromosome can be split into K loops and each of which represents the driving route of an EDV. A chromosome code is assigned to three EDVs and delivery routes are shown in Fig. 2, where $1, 2, \ldots, 13$ represent ESs, 14 represents DC and 15, 16 represent interruption points.

Selection Operation: At the beginning of GA, some particularly excellent individuals are usually produced. If the optimal individuals are selected according to the fitness value from high to low or according to the proportion, these abnormal individuals will control the selection process because their competitiveness is too prominent. As a result, at the later stage of evolution, these abnormal individuals occupy the whole population and their potential for further optimization is reduced. In the end, we may not get the global optimal solution but only get the local optimal solution. Hence, we use the Tournament Selection to select individuals by randomly selecting $n$ individuals from the entire population (usually $n = 2$) and pitting them against each other in a series of competitions, leaving only the best of them. This method doesn't need to sort all the fitness values, is not easy to fall into the local optimum and has a small time complexity (*Shukla, Pandey & Mehrotra, 2015*).

Crossover Operation: GA crossover operation exchanges some genes between two chromosomes in some way to produce a new superior individual. In this paper, Partial-Mapped Crossover (PMX) (*Goldberg & Lingle, 1985*) combined with Order Crossover (OX) is used, where the probability of executing PMX is $\delta$ and the probability of executing OX is 1-$\delta$. Both methods involve setting up two intersections, intercepting the gene fragments between the two intersections and recombining the genes after the crossover. After the crossover operation, it can better enhance the ability of GA to open up new search space, so that the final obtained offspring will be closer to the optimal solution of the problem.

Mutation Operation: The mutation operation can maintain the diversity of the population to some extent, increase the local search ability of the algorithm and prevent the premature convergence of the algorithm. The mutation operations adopted in this paper include exchange mutation, inversion mutation and insertion mutation. Here we

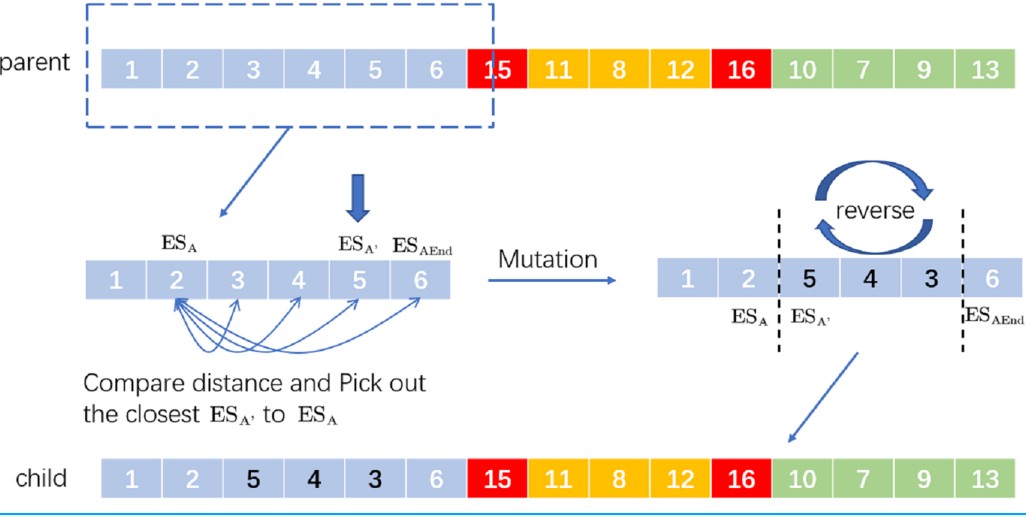

**Figure 3 An example of selective inversion mutation.**

transform inversion mutation and insertion mutation into selective inversion mutation (SIM) and selective insertion mutation (SIM') to change the search ability of the whole algorithm, where the probability of executing SIM is $\mu$ and the probability of executing SIM' is $\sigma$.

Inversion mutation refers to the 180-degree turnover of a gene segment in the chromosome. When it is applied to VRP, it means to reverse the driving route from $ES_A$ to $ES_B$ to produce a new distribution route. For the operation of SIM, randomly select a $ES_A$ then find the last $ES_{AEnd}$ in the distribution route where the $ES$ is located. Between $ES_A$ and $ES_{AEnd}$, select a location nearest to $ES_A$ and then reverse the whole distribution route from $ES_{ARight}$ to $ES_{A'}$ to generate a new distribution route. For example, the process of SIM is in Fig. 3.

We randomly select a $ES_2$ and we know $ES_6$ is the last station of the distribution route according to interruption point 15. If the distance between $ES_5$ and $ES_2$ is the shortest, the new delivery route after selective inversion mutation is 'child'.

On the whole, selective inversion mutation can find the good evolutionary characteristics of the gene sequence without disrupting the good chromosome sequence.

Insertion mutation refers to randomly taking two positions on the chromosome and then the second gene is inserted after the first one. Corresponding to the VRP, $ES_B$ is taken as the next driving destination of $ES_A$. The newly proposed SIM' is to randomly select two ESs ($ES_A$, $ES_B$) and then find the last $ES_{BEnd}$ in the distribution route where $ES_B$ is located. Between $ES_B$ and $ES_{BEnd}$, select a $ES_A$', which is closest to $ES_A$ and compare the distance between them and the distance between $ES_A$ and $ES_{ARight}$. If the former is less than the latter, remove A' and insert it after A to form a new distribution route; otherwise, the original distribution route will not be changed. For example, the process of SIM' is in Fig. 4.

We randomly select two ESs: $ES_4$ and $ES_{10}$. Behind $ES_{10}$ (including $ES_{10}$), we search which station is closest to $ES_4$. If the distance from $ES_9$ to $ES_4$ is closest and less than the distance from $ES_4$ to $ES_5$, the new delivery route after selective insertion mutation is 'child'.

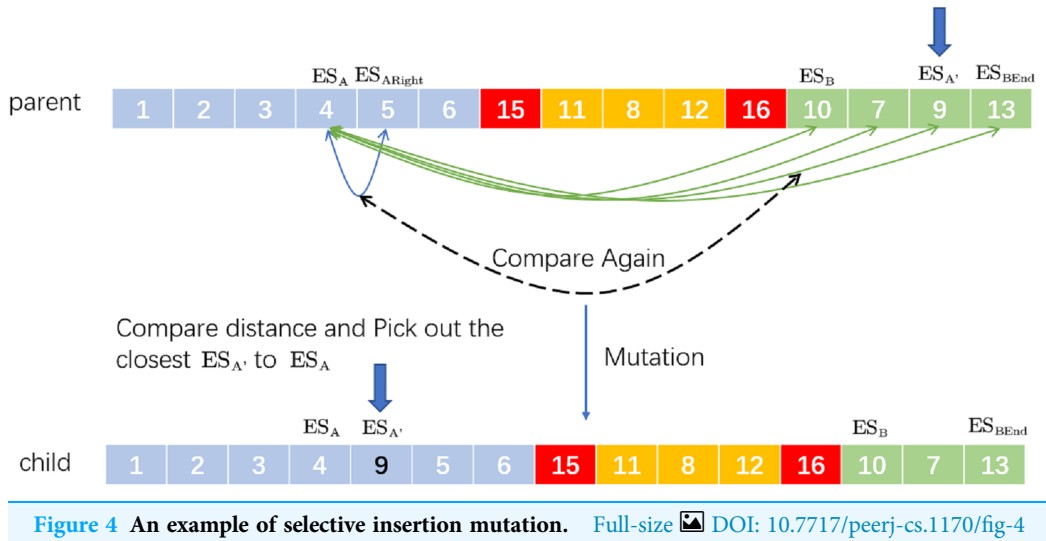

**Figure 4 An example of selective insertion mutation.**

This method expands the diversity of chromosomes to a certain extent and helps to find more effective solutions.

Next we will introduce each sub model in the multistage model in turn. The first model is centered on the farthest distance traveled by the EDV to get the ESs that EDV needs to pass through and then gives a preliminary driving route. According to the route of each EDV, each vehicle will pick up the packages to be delivered at DC. The second model is based on the first model to carry out detailed static route planning for each EDV and work out a more reasonable route to complete delivery. The third model is to obtain dynamic traffic congestion to carry out route planning, develop a more time-saving route scheme and effectively avoid the delay of delivery due to entering the congested road section.

## The application of improved GA to multistage model

Improved GA with clustering algorithm for package arrangement Here, in order to solve the delivery problem in a large area efficiently, the improved GA needs to generate a better initial population and complete path planning more quickly. Considering that ESs with similar locations can be arranged to the same EDV. We employ k-means++ clustering algorithm (*Arthur & Vassilvitskii, 2006*) to enhance the searching ability of the improved GA called "KGA-based allocation algorithm". This algorithm can group stations with similar locations and generate better individuals, which make the package arrangement more excellent in accuracy and speed.

Under complex requirements, the GA usually adopts the scheduling strategy and objective that can satisfy all demands as soon as possible. Therefore, we choose to minimize the driving distance and minimize the gap between the actual driving distance and the maximum driving distance of the EDV. The purpose is to evenly distribute delivery routes of different lengths to each EDV, *i.e.*, the longer delivery route is allocated to EDV with longer driving distance. So we transform the multi-objective optimization

---

**Algorithm 1 KGA-based allocation algorithm.**

Input: *d, edvd, P, c, c*_rate, *m*_rate.

Output: *TDD, DR, dd$_k$, k* ∈ 1, 2, . . . , *K*.

for i in range P:

    Initialize the initial population by K-means++.

j=1

while j < I:

    selection: select P excellent individuals by tournament selection.

    crossover: cross two individuals with PMX or OX to generate new individuals when p <= c_rate.

    mutation: mutate the individual with SIM or SIM' when p <= m_rate.

    Gets the distance traveled by all individuals.

    fitness evaluation: calculate the fitness of all individuals in the population.

    Record the optimal individuals of each generation.

    if fitness of the optimal individual in each generation is greater than the global optimal individual.

        Update the information of the global optimal individual.

    for m in range len(edvd):

        if bd[m]>edvd[m] then flag=False.

    if (flag == False) then j++.

    else I=j and return *TDD, DR, dd$_k$*.

    Update the population.

---

problem into a single-objective optimization problem, whose objective function is defined as follows:

$$\min Z = \omega_1 * \sum_{k=1}^{K} dd_k + \omega_2 * \sum_{k=1}^{K} (D_k - dd_k) \tag{9}$$

where $\omega_1$ can be called as the distance weight, $\omega_2$ can be called as the "gap" weight and $\omega_3$ can be called as time weight appearing in below. The KGA-based allocation algorithm can quickly obtain the ESs that each EDV should passes through and then we can arrange the delivery for each EDV.

Algorithm 1 shows the pseudocode of KGA-based allocation algorithm.

Where I is the maximum number of iteration, P is the size of population, c_rate and m_rate are the crossover and mutation rates, TDD is total driving distance, *i.e.*, sum of distances traveled by five EDVs, bd[m] is the distance that the m-th EDV in the global optimal individual needs to drive and DR is the driving route of all EDVs. So far, the first stage of express delivery is completed and the detailed route planning will be realized by the second and third stages.

Improved GA for static route planning: In the phase of solving the static route planning model, stations that each EDV needs to pass through have been obtained. Next, we have to

decide the routing plan for each EDV with assigned stations. Hence clustering algorithm is no longer used for population initialization. We take the driving routes of each EDV generated in the previous stage and shuffle the sequence of the routes to form a new individual. $P$ individuals are combined into a new population, where each individual is represented as a different driving route of the EDV. It should be noted that $d_{ij}$ is the distance from the $ES_i$ to the $ES_j$ and the driving speed of vehicles on the road is not only affected by road speed limit but also by traffic congestion or special weather. Therefore, we introduce the traffic congestion coefficient $c$ to estimate the driving time (DT) taken by vehicles from station $i$ to station $j$, and the formula is as follows:

$$DT_k = \sum_{i=1}^{N+1} \sum_{j=1}^{N+1} \left( c_{ij} * \frac{x_{ijk} * \mathrm{d}_{ij}}{v} \right), k \in \{1, 2, \ldots, K\} \tag{10}$$

We mainly focus on the urban logistic distribution, where $v$ is the driving speed of EDV in the process of distribution in the city, $c$ indicates the congestion coefficient of the road between different ESs and $c \in [1, 5]$. The larger value of $c$ is, the more serious the road congestion is and the calculated driving time is greater than the time that it drives at a constant speed $v$. For this circumstance, the routing plan should take the shortest driving time as the objective condition that EDV's driving distance may be more longer for preventing driving into congested areas and ensuring that each EDV can spend the least time to complete delivery. Therefore, we also transform the multi-objective problem into a single-objective optimization problem in order to shorten the traveling time and distance at the same time and its objective function is defined as follows.

$$\min Z = \omega_1 * dd_k + \omega_3 * \mathrm{DT}_k \tag{11}$$

The improved GA is used to solve the static route planning model, the pseudocode of GA-based static route planning algorithm is shown in Algorithm 2.

Where *pop* represents the whole population. This algorithm obtains the traffic congestion coefficient between each ES in the transportation network after the vehicles complete packages loading and then makes the route planning. Once the route of each EDV is determined, it will not be changed.

Improved GA for dynamic route planning: According to solving the static route planning model, a route planning is obtained for each EDV. However, road information collected at a certain time point is time-sensitive. If we only follow the distribution route obtained before departure, it's likely to cause vehicles to enter the congested road and delay the delivery time. Therefore, we want to establish dynamic route planning according to real-time traffic conditions. In order to search the distribution route dynamically, we need to consider how to deal with the dynamic model. Generally, there are two dynamic calculation methods. One is to calculate at a fixed time interval. Starting from the departure of ESs, every time interval, the system will automatically obtain the traffic information between stations in the transportation network and re-plan the route. The characteristics of this method are as follows: when the time interval is too large, the traffic

**Algorithm 2 GA-based static route planning algorithm.**

Input: *d, edvd, P, c, DR, c_rate, m_rate.*

Output: *TDD, DR, $dd_k$, $k \in 1, 2, \ldots, K$.*

for i in range(len(DR)):

    pop = [].

    pop.append(DR[i]).

    for k in range(P):

        shuffle DR[i] and add it to population.

    Take pop[0] as the global optimal individual and calculate its fitness.

    j=1.

    while j < I:

        selection, crossover and mutation.

        Obtain driving distances of all individuals and calculate their times according to *c*.

        fitness evaluation: calculate fitness of individuals of population.

        Record the optimal individual of each generation.

        if fitness of the optimal individual in each generation is greater than the global optimal individual.

            Update the information of the global optimal individual.

        Update the population.

        j++.

    return *TDD, DR, $dd_k$.*

information will be out of date; when the time interval is too short, the route planning will be more accurate but the calculation time will be longer.

The other method is to calculate at the node. After arriving at any node, the EDV will re-obtain the traffic information in the transportation network to re-planing the distribution route according to the remaining nodes until it finally returns to the DC. Here, we adopt the second method and make corresponding improvements.

When EDV arrives at a certain $ES_i$ and finishes unloading, it still follows the distribution route plan made at the previous node: drive to $ES_j$. At the same time, according to the comparison between $c_{ij}$ (represents the traffic congestion coefficient between $ES_i$ and $ES_j$) and $\beta$ (the threshold determined by experience), if $c_{ij} < \beta$, it means that the congestion between the two stations is not very serious and EDV can still drive according to the distribution route, *i.e.*, from $ES_i$ to $ES_j$. Otherwise, it is necessary to re-planing the distribution route for the remaining distribution nodes. At this circumstance, according to the new distribution route, the EDV should drive from $ES_i$ to $ES_k$ (other node with a lower congestion coefficient). The objective function of this model is consistent with the static route planning model.

The improved GA is used to solve the dynamic route planning model, the pseudocode of GA-based static route planning algorithm is shown in Algorithm 3.

**Algorithm 3  GA-based dynamic route planning algorithm.**

Input: $d$, $edvd$, $P$, $c$, $DR$, $c\_rate$, $m\_rate$;

Output: $TDD$, $DR$, $dd_k$, $k \in 1, 2, \ldots, K$;

for i in range(len(DR)):

  chrome=DR[i].

  for k in range(len(chrome)):

    Obtain real-time road congestion coefficient matrix: conMatrix.

    if (conMatrix[chrome[k]-1,chrome[k+1]-1]<= $\beta$):

      continue.

    pop = [ ].

    Initialize the population.

    j=1

    while j <= (len(chrom)-k) * 2:

      selection, crossover and mutation

      Obtain driving distances of all individuals and calculate their driving times according to $c$.

      fitness evaluation: calculate the fitness of all individuals in the population.

      if fitness of the optimal individual in each generation is greater than the global optimal.

        Update the information of the global optimal individual.

      Update the population.

      j++

    obtain the next stop of the $i - th$ EDV.

  obtain the $DR_i$ of the i-th EDV.

return $TDD$, $DR$, $dd_k$

Where chrome represents an individual in the population. This algorithm re-plans the route when the vehicles encounter serious traffic jams until each EDV completes the delivery.

## EXPERIMENTS AND RESULTS

In this section, we will evaluate the performance of the integrated approach that includes both the proposed multistage model and the improved GA. For the former, we mainly evaluate it according to the total driving distance and driving time of EDVs, while for the latter, we mainly evaluate it according to the average number of iterations of the algorithm. We conduct a series of experiments based on a dataset collected from a real-world delivery services. The details of dataset is shown in Table 2. There is one DC for sorting and temporarily storage packages and 1,000 ESs randomly around the DC. Five EDVs are assigned to the DC for delivery. We assume that the average driving speed of EDV in the city is 30 km/h and the maximum driving distance of these five EDVs is different, which are 2,000, 2,200, 1,900, 2,400 and 1,800. We take the threshold of traffic congestion

**Table 2 Details of synthesized data set.**

| Data item | Value |
|---|---|
| The number of ES | 1,000 |
| The number of DC | 1 |
| The number of EDV | 5 |
| Average speed of EDV | 30 km/h |
| Threshold of traffic congestion coefficient | 3 |
| Maximum driving distance of five EDVs | [2,000, 2,200, 1,900, 2,400, 1,800] |

coefficient as three, which means that when the congestion coefficient of a road is greater than the threshold value, path planning will be redone.

Three experiments were implemented using Python language on personal notebook: AMD Ryzen 7 4800U with Radeon Graphics (1,800 MHz) @16 GB memory and 64-bit. For each experiment, we analyze the results according to the total distance and driving time of EDVs in each model. Meanwhile, finding the best solution in finite iterations is also a main criterion for evaluating the effectiveness of the algorithm. We did the following experiments mainly according to each stage of the multistage model in order to better reflect the progressive relationship between each layer.

In the first experiment, we compare the performance difference between ordinary genetic algorithm and improved genetic algorithm in route allocation. Given the maximum distance of each EDV, we compare which method can allocate the driving route of each EDV in the shortest time (the least number of iterations). Due to the different EDVs have different maximum driving distance, the allocated area blocks (clusters classified according to the clustering algorithm) should also be different. The result is shown in Fig. 5.

As shown in the figure, under the condition of the maximum driving distance of five EDVs: 2,000, 2,200, 1,900, 2,400 and 1,800, the results of one experiment are shown as follows: 70 iterations based on the improved genetic algorithm can meet the requirements of the condition, while it takes 202 iterations based on the traditional genetic algorithm to determine the delivery stations for each EDV. According to the experimental results, the average time of the first stage based on the improved genetic algorithm is 12.14 s, while the average time based on the traditional genetic algorithm is 41.3 s. The traditional genetic algorithm only uses OX for exchange operator, insertion mutation and inversion mutation for mutation operator, but it's also based on k-means++ algorithm. If the traditional GA is used alone, thousands of iterations are required. It can be seen that the improved genetic algorithm has better convergence effect.

In the second experiment, we optimize the allocation route derived from the improved GA in experiment 1 based on the traditional VRP model. This experiment proves that the algorithm we use can optimize the distribution route well and ensure that the driving distance of each EDV is as short as possible under the shortest driving time. Optimized routes are shown in Fig. 6.
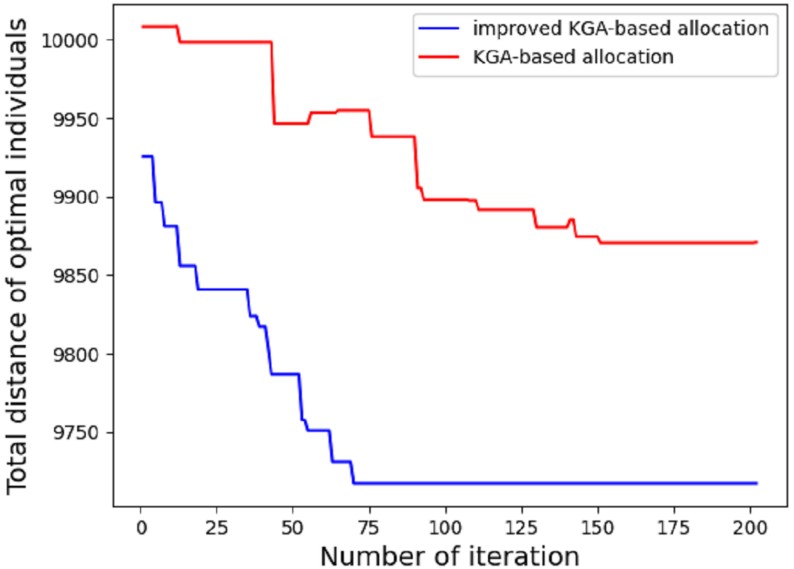

**Figure 5   Total distance of different algorithm.**

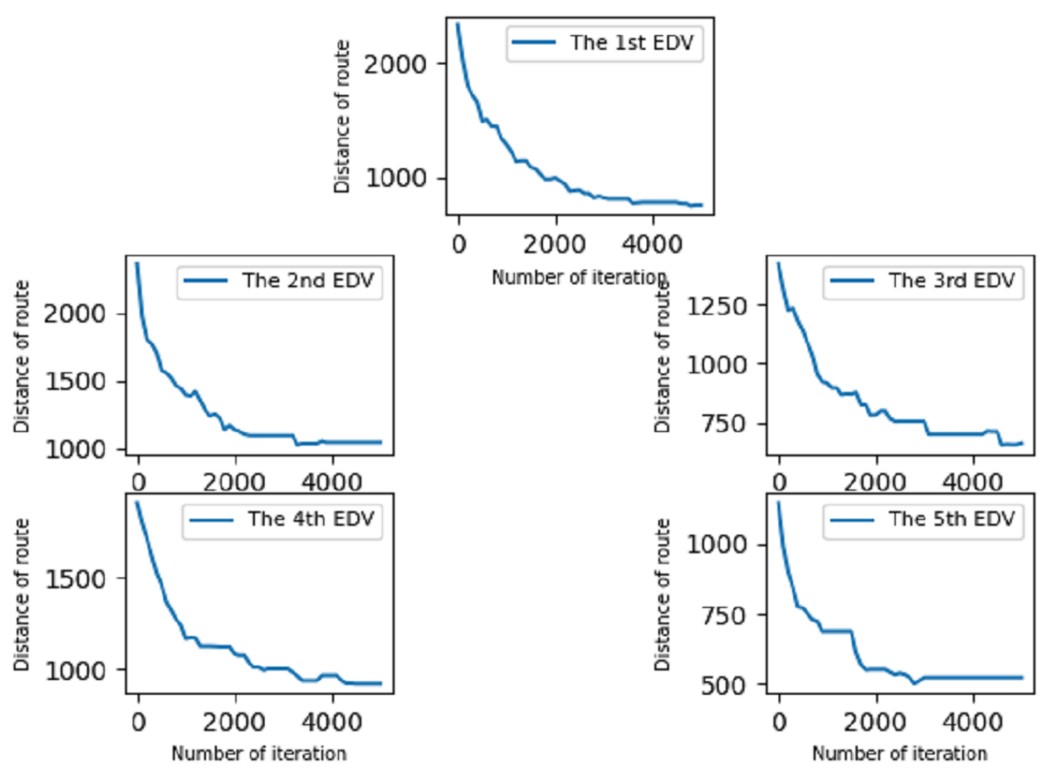

**Figure 6   Optimization of different distribution routes.**

As shown in Fig. 6, route optimization was carried out for each EDV and the delivery distance was greatly reduced. Compared with the total driving distance of each EDV in experiment 1, the final total driving distance was reduced by 66.7% by optimizing the

**Table 3 Settings of object function and algorithm.**

| Variable | Value |
|---|---|
| Objective function | |
| $\omega_1$, $\omega_2$ and $\omega_3$ | 1, 1,500 and 250 |
| Improved Genetic algorithm | |
| Mutation rate | 0.25 |
| crossover rate | 0.8 |
| population size $P$ | 30 |
| PMX rate $\delta$ | 0.5 |
| SIM and SIM' rate $\mu$, $\sigma$ | 0.5, 0.5 |
| maximum iteration number $I$ | 1,000 and 5,000 |

delivery route of each EDV in experiment 2. The parameters of the objective function and improved genetic algorithm are shown in Table 3.

In the second experiment, we performed 5,000 iterations for each EDV route update, taking an average of 230 s. However, if the experiment is done on a computer with higher computing power, it is believed that the time will be greatly reduced. Since the algorithm has a good effect in static model, so we continue to study the effect of the algorithm in dynamic model.

In the third experiment, we adopt dynamic VRP model to deal with changing traffic conditions. The route planning shall be carried out again after the vehicles reach each ES and the final results (delivery time and delivery distance) shall be compared with the results obtained from the static route planning.

In order to calculate the delivery time under the static model (HFVRP+SVRP), we use the delivery route of each EDV obtained in experiment 2. Because the delivery route has been fixed, so we only need to obtain the traffic congestion coefficient (We use random numbers to replace the traffic congestion coefficient obtained by GPS) between the $ES_A$ and the next $ES_B$ after the EDV arrives at $ES_A$ and then calculate the time required to reach the next $ES_B$. Finally, we add up each section of driving time to get the total delivery time. The time calculation of dynamic model (HFVRP+SVRP+DVRP) also accumulates the driving time between the two ESs, which is similar to the static model, but the distribution path will be adjusted dynamically. In addition, we calculate the driving distance of each EDV in the two models and analyze it with the delivery time. By comparing delivery distance and time, the results of dynamic model have both reduced in some different degree. Specifically, the delivery distance has an average decrease of 7.54% and delivery time declines 19.9%. These results are shown in Table 4.

Because the DC receives the packages frequently, in order not to pile too many packages, it is necessary to arrange new delivery in time and deliver the packages to the customers as early as possible. Therefore, we prefer EDV to finish each delivery earlier, so as to start the next delivery earlier. It can be seen from the results that the driving time decreased significantly and the driving distance as a whole shows a downward trend. This

**Table 4 Experimental results.**

| Lines | Driving distance | | | Estimation of driving time | | |
|---|---|---|---|---|---|---|
| | Static model | Dynamic model | Proportion of reduce | Static model | Dynamic model | Proportion of reduce |
| Distribution line1 | 541.4 | 484.6 | 10.5% | 51.7 | 43.9 | 15.1% |
| Distribution line2 | 722.6 | 735.1 | – | 72.7 | 63.9 | 12.1% |
| Distribution line3 | 599.7 | 617.4 | – | 62.1 | 53.7 | 13.5% |
| Distribution line4 | 913.9 | 717.4 | 21.5% | 91.6 | 68.4 | 25.3% |
| Distribution line5 | 547.1 | 405.4 | 25.9% | 55.0 | 36.6 | 33.5% |

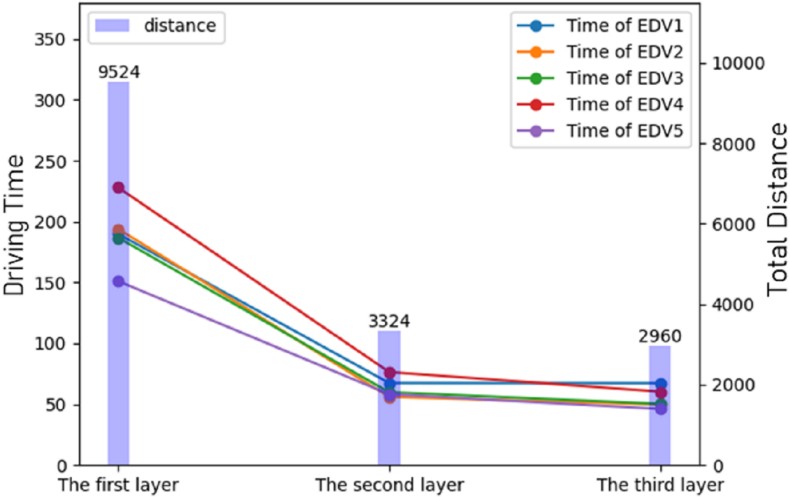

**Figure 7 The result of optimization of different stages.**

shows that the current algorithm will still fall into local optimization when there are too many data points and a small number of iterations, so we apply the algorithm to the dynamic model to to obtain the optimal solution by continuously reducing the size of the dataset. In the third stage, there are only few factors that need to consider, which makes it easier to achieve global optimization on the basis of the optimized route when re-planning the route. Obviously, the results can meet our goal, that is to complete delivery in the shortest time.

At the end of the experiments, all the results were summarized and the results of the three stages of route planning were displayed. It can be seen that after route planning, the delivery route was greatly reduced and the delivery time was also greatly reduced, as shown in Fig. 7.

## CONCLUSIONS

With the rapid development of the logistics industry and the the rise of online commerce, using DVDS is a promising way to improve delivery efficiency and users' satisfaction. In this article, we propose an integrated approach with a multistage model and an improved genetic algorithm to solve the problem of distribution route planning for DVDS. To be

specific, the model in the first stage was used to determine the package arrangement, then a preliminary distribution route was generated in the second stage and finally the distribution route plan for each EDV was optimized in the third stage considering the traffic congestion situation. In short, the goal of this study was to improve delivery efficiency with driverless vehicles, *i.e.*, to complete the delivery in the shortest time. Experimental results have shown that our proposal can improve delivery efficiency. This work may provide some practical guidance and valuable insights for designers of delivery services.

In the future work, we will consider the utilization of real-time GPS information, improve the efficiency and performance of the algorithm, and achieve more accurate route planning.

### Funding
This work was supported by the Scientific Research Funds of Northeast Electric Power University (No. BSZT07202107). The funders had no role in study design, data collection and analysis, decision to publish, or preparation of the manuscript.

### Grant Disclosures
The following grant information was disclosed by the authors:
Scientific Research Funds of Northeast Electric Power University: BSZT07202107.

### Competing Interests
Yanling Wu is employed by Hebei Hanguang Industy Co., Ltd.

### Author Contributions
- Tianyang Li conceived and designed the experiments, analyzed the data, performed the computation work, authored or reviewed drafts of the article, and approved the final draft.
- Zhangyi He performed the experiments, analyzed the data, performed the computation work, prepared figures and/or tables, and approved the final draft.
- Yanling Wu analyzed the data, prepared figures and/or tables, and approved the final draft.

### Data Availability
The data is available at Zenodo: Zhangyi-He1207. (2022). Zhangyi-He1207/Route-Planning-Algorithmfor-Driverless-Vehicle: V1.0.0 (V1.0.0). Zenodo. https://doi.org/10.5281/zenodo.7046194.

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
