# Peer review of "An integrated route planning approach for driverless vehicle delivery system"

_PeerJ Computer Science, doi:10.7717/peerj-cs.1170_

## Round 0.1 · original submission · Major Revisions

Based on the comments from reviewers, the paper needs to be improved through a major revision. The proposed method is presented in the GA framework, however the performance comparison is not sufficient. More experimental studies are expected and the typos in the manuscript should be corrected before re-submission.

Reviewer 1 ·

Basic reporting

In this paper, the authors propose an approach to generating efficient route plans for autonomous delivery vehicles by considering not only the delivery routes but also other related factors such as the remnant of battery power, weather, and traffic conditions. They propose a generic algorithm based multi-stage optimization algorithm and evaluate their approach by using a real-world dataset.
The research topic is meaningful and the paper is well organized, while some details need to be further revised:
The pictures in the text are extremely unclear.

Experimental design

The experimental part needs to be reconstructed, and the dataset and evaluation metric should also be described in detail.
The authors are suggested to explain the meaning of the multi-stage model and how it works in the introduction briefly.
The detailed time consumption of the algorithm is not given.

Validity of the findings

1. In the Introduction, previous methods about route plans for autonomous delivery vehicles should be reorganized and summarized, and then state the existing problems.
2. In the Introduction, the description of the literature is briefly mentioned, and the advantages and disadvantages of the algorithms in the literature need to be added.

Reviewer 2 ·

Basic reporting

In order to address the challenge of routes planning for driverless vehicles, such as load capabilities,
power limits and traffic conditions, this manuscript tried to integrate multistage model with improved genetic algorithm to obtain the optimal delivery. It is important for driverless vehicles and I believe this work will help to accelerate express delivery service efficiency. The manuscript appears well written but there are on numerous occasions misprints and grammar mistakes that are distracting. The manuscript should be revised on that aspect.
1. Line 43 should be rewritten “providing an effective delivery path”, not “a effective”.
Also in line 44, what is “salving cost”, do you mean “saving cost” ?
2. From line 47 to line 61. Firstly, the cited literature (Golden, 1984) for traditional method is too old to show the advancement of the manuscript. And, in the part of related works, the literature of HFVRP model is up-to-date. Secondly, why “the traditional model lacks the expression of these crucial factors”, the content of this part is not clear. Thirdly, why the method of VRP and DVRP without any citations. Sufficient introduction and background are needed.
3. In the part of related work, the authors need provide a more comprehensive literature review on the algorithms in this field, rather than describe the advantages of driverless vehicles.

Experimental design

In the part of experimental design, the submission clearly defined the research question, including definition of entities, multistage model construction and hypothesis, and problem formulation. According to the supplementary materials provided, the submission could be described with sufficient information to be reproducible and reproducibility.
1. There are many solutions to solve the problem of route planning. What are the advantages and advancements of manuscript experimental design? The authors should explain more about the novelty of the experimental design used in this manuscript.
2. The objective function and constraints of the optimization problem is unclear - The authors should describe the formulation in the materials and method clearly and thoroughly.

Validity of the findings

Through three experiments in an urban scenario with a realistic delivery service show the validity of the proposition. But before publication, there are still some concerns needed to be solved.
1. Genetic algorithm is a kind of biologically inspired computing. What is the superiority of proposed improved genetic algorithm compared with other inspired computing algorithms.
2. The 'Conclusion' section of the manuscript is too general. It must give brief account of the major findings obtained in the study.

Additional comments

Based on designed multistage model and improved genetic algorithm, this manuscript tried to provide insights for delivery industry and improve delivery efficiency with driverless vehicles. The manuscript can be accepted for publication if it will be scientifically edited to follow strictly instructions of the journal.

Reviewer 3 ·

Basic reporting

This paper presents a route planning problem for electric driverless vehicles (EDV) to solve the transportation delivery problem in the express delivery industry for short-distance logistics. By considering various problems that EDV may encounter in operation, such as load capacity, power limitation and traffic conditions. They use multi-stage modeling to cope with the changing traffic environment. An improved genetic algorithm is used to complete the selection of the best delivery schedule.
The issues discussed in this paper seem timely and important, from the testing results of the authors, their approach showed some performance enhancements. However, there are some problems in the manuscript that need to be revised.
1.The contributions explained in the introduction section are simply a summary of the proposed approach.
2.What are the unsolved problems or the limitations of the existing approaches that the proposed approach can now effectively deal with?
3.What are the relationships among the multistage model and traditional VRP models, such as HFVRP and DVRP?

Experimental design

The experimental design of this paper reaches the desired levels of soundness. However, the objective assessments of their approach as well as analysis of their approach for obtaining better results are missing.

Validity of the findings

The findings of this manuscript have the desired levels of verifiability based on an available replication package containing the data and software related with the paper for interested readers. While, a brief description of current limitations of proposed algorithm, since any experiment/case study is subject to such threads, would be welcome.

---

## Round 0.2 · accepted · Accept

Based on the reviewers' comments, all the concerns have been addressed well in the revised version. Therefore, I believe that the current manuscript is ready for publication.

Reviewer 1 ·

Basic reporting

no comment

Experimental design

no comment

Validity of the findings

no comment

Additional comments

The authors have edited the manuscript to address my concerns. I have no further comments.

Reviewer 2 ·

Basic reporting

no comment

Experimental design

no comment

Validity of the findings

no comment

Additional comments

Based on designed multistage model and improved genetic algorithm, this manuscript tried to provide insights for delivery industry and improve delivery efficiency with driverless vehicles. The manuscript has been revised as required, and meets the PeerJ criteria and should be accepted as is.